

# A new comprehensive approach for regional drought monitoring

Rizwan Niaz[1], Mohammed M. A. Almazah[2,3], Ijaz Hussain[1], Muhammad Faisal[4], A. Y. Al-Rezami[5,6] and Mohammed A. Naser[2]

[1] Department of Statistics, Quaid-i-Azam University, Islamabad, Punjab, Pakistan
[2] Department of Mathematics, College of Sciences and Arts (Muhyil), King Khalid University, Muhyil, Saudi Arabia
[3] Department of Mathematics and Computer, College of Sciences, Ibb University, Ibb, Yemen
[4] Faculty of Health Studies, University of Bradford, Bradford, UK
[5] Department of Statistics and Information, Sana'a University, Sana'a, Yemen
[6] Mathematics Department, Prince Sattam Bin Abdulaziz University, Saudi Arabia, Saudi Arabia

Corresponding authors
Rizwan Niaz,
razwanniaz11@gmail.com
Mohammed M. A. Almazah,
mmalmazah@kku.edu.sa

## ABSTRACT

The Standardized Precipitation Index (SPI) is a vital component of meteorological drought. Several researchers have been using SPI in their studies to develop new methodologies for drought assessment, monitoring, and forecasting. However, it is challenging for SPI to provide quick and comprehensive information about precipitation deficits and drought probability in a homogenous environment. This study proposes a Regional Intensive Continuous Drought Probability Monitoring System (RICDPMS) for obtaining quick and comprehensive information regarding the drought probability and the temporal evolution of the droughts at the regional level. The RICDPMS is based on Monte Carlo Feature Selection (MCFS), steady-state probabilities, and copulas functions. The MCFS is used for selecting more important stations for the analysis. The main purpose of employing MCFS in certain stations is to minimize the time and resources. The use of MCSF makes RICDPMS efficient for drought monitoring in the selected region. Further, the steady-state probabilities are used to calculate regional precipitation thresholds for selected drought intensities, and bivariate copulas are used for modeling complicated dependence structures as persisting between precipitation at varying time intervals. The RICDPMS is validated on the data collected from six meteorological locations (stations) of the northern area of Pakistan. It is observed that the RICDPMS can monitor the regional drought and provide a better quantitative way to analyze deficits with varying drought intensities in the region. Further, the RICDPMS may be used for drought monitoring and mitigation policies.

# INTRODUCTION

Drought can be described as a recurring and challenging natural calamity that arises when precipitation is substantially less than its average time (*Smucker & Wisner, 2008*; *Gupta, Tyagi & Sehgal, 2011*; *Caine et al., 2019*; *Van Loon, 2015*; *SafarianZengir,*

**PeerJ** _______________________________________________________

_Sobhani & Asghari, 2020_; _Ndlovu & Mjimba, 2021_; _Brakkee, van Huijgevoort & Bartholomeus, 2022_). The occurrence of drought can impact the economy, environment, and social stances (_Hlavinka et al., 2009_; _Ding, Hayes & Widhalm, 2011_; _Van Dijk et al., 2013_; _Sahoo et al., 2015_; _Ndlovu & Mjimba, 2021_; _Alsafadi et al., 2022_). Drought exhibits extensive spatial and temporal discrepancy in different climatic conditions and localities. Numerous researchers have developed several methods and techniques for studying the spatial and temporal variation of drought occurrences (_Smucker & Wisner, 2008_; _Schwantes, Swenson & Jackson, 2016_; _Rahman & Dawood, 2018_). However, drought can be considered a highly inconsistent complex phenomenon and challenging to understand its onset and termination phases (_Arabameri et al., 2021_; _Savari, Damaneh & Damaneh, 2022_). Moreover, the climatology community categorizes drought into four types to differentiate its special effect: socio-economic; hydrological; meteorological, and agricultural (_Duan & Mei, 2014_; _Huang et al., 2015_; _Martínez-Fernández et al., 2015_; _Javed et al., 2021_; _Wang et al., 2022_). The standardized procedures are frequently used for drought monitoring, evaluation, and characterization (_Wang et al., 2016_; _Tirivarombo, Osupile & Eliasson, 2018_; _Wang & Yu, 2021_).

In literature, various Standardized Drought Indices (SDIs) have been proposed to characterize and monitor drought occurrences (_Erhardt & Czado, 2015_; _Erhardt & Czado, 2018_; _Neeti et al., 2021_; _Wang & Yu, 2021_). Among these SDIs, the index most widely employed in various studies and applications is the Standardized Precipitation Index (SPI) (_McKee, Doesken & Kleist, 1993_). Further, in 2009, World Meteorological Organization (WMO) has strongly recommended the SPI for assessing and monitoring meteorological drought. Subsequently, the SPI has been frequently used to calculate the precipitation inadequacy for varying time scales to quantify meteorological drought in numerous studies (_Sigdel & Ikeda, 2010_; _Chen & Sun, 2015_; _Schubert et al., 2016_). The SPI can be applied to assess droughts in the different climatic zones (Keyantash & Dracup, 2002; _Zhou & Liu, 2016_). Moreover, the index becomes challenging when required expeditiously knowledge about the precipitation shortages and drought probability in the homogenous environment where the characteristics over the region produce similar results (_Niaz et al., 2020b_). Therefore, new methods and techniques are needed to obtain comprehensive information in particular climatic scenarios.

Often the univariate-based frameworks and methods are utilized for the identification and description of the drought occurrences. However, several characteristics in hydrological phenomena have convoluted dependence structures which cannot be described explicitly in the univariate setting and lead to generating poor outcomes (_Grimaldi & Serinaldi, 2006_; _Zhang, Su & Feng, 2021_). Therefore, to improve the efficiency of the estimates, multivariate techniques are used in several analyses to assess the dependence structures among the hydrological characteristics (_Nagaraju et al., 2017_; _Atazadeh et al., 2021_). These techniques consider bivariate probability distributions, such as bivariate exponential, gamma, Gaussian, and extreme value distributions. In such techniques, the critical weakness arises because the substantial details more than bivariate cases are not described explicitly. Further, these bivariate cases are formed with the constraint that marginal must have the same probability distribution. Therefore, it is

required to utilize those techniques which more explicitly describe and separately define the dependence structures to overcome these concerns.

Therefore, adaptability exists in the modeling of varying dependence structures, that underpins the use of copulas functions, which have vast appendages for assessing dependence structures of the multivariate processes. The copulas are used for evaluating joint distributions through distinctly classifying the dependence structures of random variables from their marginal distributions (*Nelsen, 2007*; *Nikoloulopoulos, Joe & Li, 2012*; *Durante, Fernandez-Sanchez & Sempi, 2013*; *Joe, 2014*). The application of copulas has been apparent in drought analysis (*Song & Singh, 2010*; *She & Xia, 2018*; *Farrokhi, Farzin & Mousavi, 2021*; *Najib, Nurdiati & Sopaheluwakan, 2021*; *Guo et al., 2022*; *Li et al., 2022*). Further, the vital application of copula extends for frequency analysis to combine various climatological characteristics (*Nabaei et al., 2019*; *Otkur et al., 2021*). In climatic regions like the northern area of Pakistan, the leading cause of drought occurrences is inadequate precipitation in the span of the rainy season. An advanced copulas-based approach is used to monitor the evolution of drought probability based on the continuous updating of rainfall occurrence. This study proposes a method that provides expeditious and comprehensive insight relating to precipitation shortages and drought probability in a homogenous environment. The proposed method is known as the Regional Intensive Continuous Drought Probability Monitoring System (RICDPMS), based on Monte Carlo Feature Selection (MCFS) technique, steady-state probabilities (SSP), and copulas functions. The RICDPMS is proposed to comprehensively model the precipitation data for various drought intensities in the selected stations. Moreover, the use of copulas' function to monitor regional drought advances, specifically for the northern area of Pakistan, makes this study innovative. Further, the RICDPMS provides valuable references for assessing, monitoring, and analyzing drought occurrences, particularly in the regional setting.

# MATERIAL AND METHODS

## Description of the study area

The natural distribution of the selected region with high altitude significantly impacts other parts of the country (*Rasul & Chaudhry, 2010*). The region becomes a comprehensive source of pervading evaporative, regulates the winds, impacts on the irrigation of the agriculture sector across the country (*Malik et al., 2012*; *Naheed, Kazmi & Rasul, 2013*), although the larger area of the country goes through with the highest temperature (*Jilani, Haq & Naseer, 2007*). Therefore, the region is selected for the analysis due to the considerable dependency of the other part of the country. The fluctuating extents due to shifting weather patterns in the season are observed in different regions within the country. Recently, global warming has influenced remarkably in several parts of the country (*Malik et al., 2012*). Its impact has been spreading worldwide, and Pakistan is not alone in facing these issues, such as increased temperature and water deficiency. Frequently, drought occurrences damage the economic sector, agriculture, and natural resources. Specifically, in Sindh (province of Pakistan), it has affected human life for the last three decades. Therefore, it is essential to make drought characteristics expeditiously

by evolving comprehensive and well-managed tools and frameworks. However, the present results will significantly enhance the capability of drought monitoring and mitigation policies.

## Data and methods

For the analysis of the current study, the time series data is assembled from the Pakistan Meteorological Department through the "Karachi Data Processing Center" (KDPC) from January 1971 to 2017 December. The six meteorological stations are selected for the analysis from the northern zones of Pakistan. The selected area of the country has some importance due to its significant climatological characteristics (*Awan, 2002*). It has become an essential cause for delivering evaporative and regulating the winds that have several impacts on the irrigation of the agriculture sectors of other regions across the country (*Malik et al., 2012*; *Naheed, Kazmi & Rasul, 2013*). In the recent past, the impacts of global warming have been spreading worldwide and influencing remarkably in various regions of Pakistan (*Malik et al., 2012*). Its consequences have increased the temperature and water deficiency.

Further, drought is one of the challenging environmental catastrophes that can distress the natural life of the community directly or indirectly (*Orimoloye, Zhou & Kalumba, 2021*; *Ndlovu & Mjimba, 2021*; *Lakzehi Moghaddam & Faryadi, 2021*; *Nasrnia & Ashktorab, 2021*). Therefore, appropriate methods of drought monitoring are required to reduce the negative impacts of drought (*Santos et al., 2013*; *Mishra, Bruno & Zilberman, 2021*; *Vogel & Kroll, 2021*; *Aryal & Zhu, 2021*; *Niaz et al., 2021e*; *Niaz et al., 2021c*). The current study focuses explicitly on regional drought monitoring. For this purpose, the MCFS technique is used to obtain comprehensive regional data set for the precipitation. The probabilities obtained from the steady-states are utilized to find the precipitation thresholds for the varying drought classes in the allocated region. Moreover, the expeditious and comprehensive information regarding precipitation deficits and drought probability will help to formulate better early warning policies for drought preparedness, mitigation, and response. The SPI has been frequently utilized for developing new methodologies for analyzing meteorological drought; however, analyzing complete data with SPI for internally similar characteristics of varying homogenous stations becomes time-consuming and costly (*Niaz et al., 2020b*; *2021b*; *2021d*). Therefore, the current study provides an innovative synthesis that can help to monitor drought characteristics more expeditiously and gives a strong signal for an early warning system at the regional level. In this regard, the most appropriate distributions associated with the temporal and spatial extents are used for calculating the thresholds of precipitation for several drought intensities.

### Regional intensive continuous drought probability monitoring system (RICDPMS)

The precipitation of the northern area (see Fig. 1) is selected for the study from January to June (six months). It has been observed that most of the rainfalls for this region occur during this period. Therefore, the selected rainy period is obtained from the collected data.
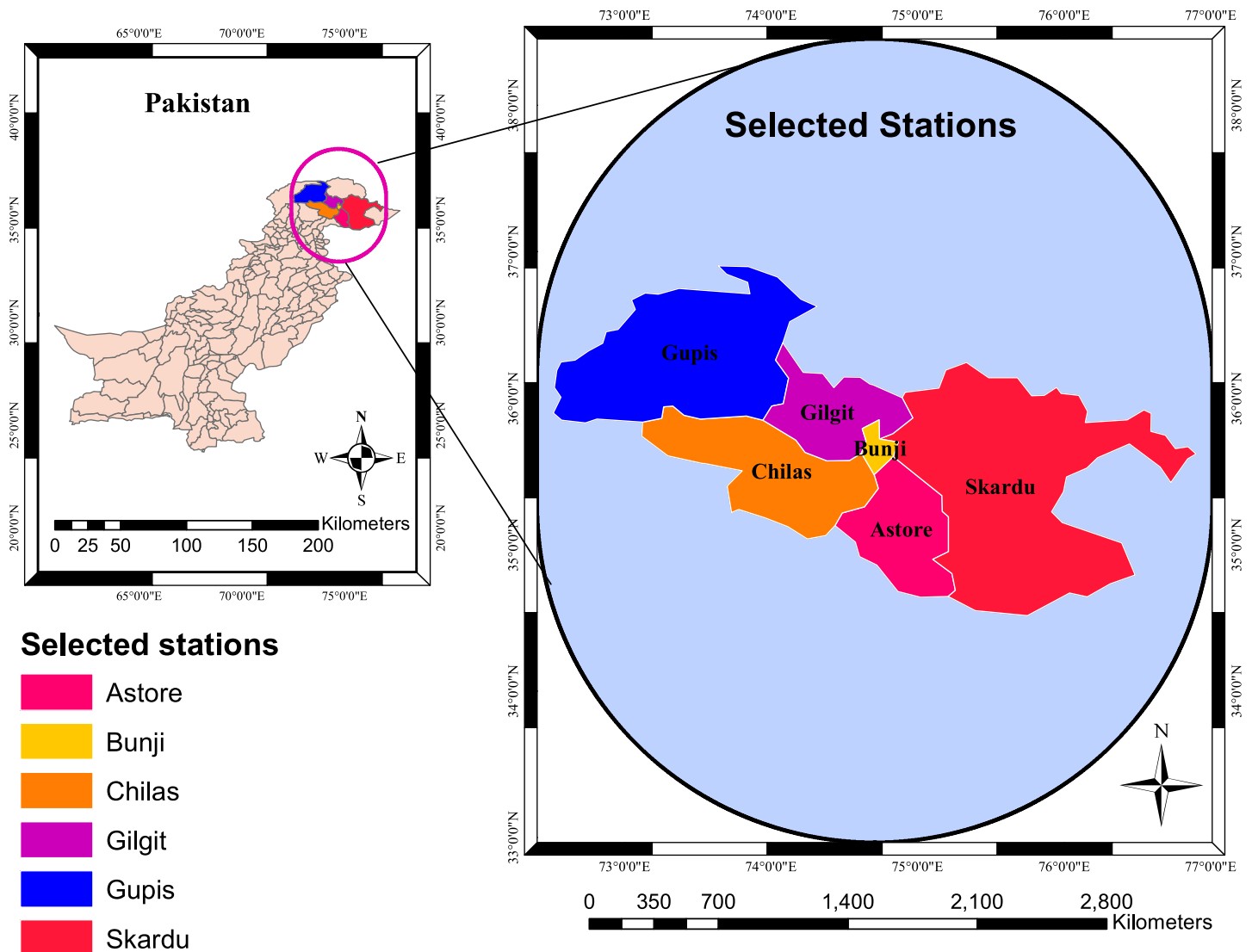

**Figure 1 The selected stations.** The six meteorological stations selected from the northern areas of Pakistan.

Further, identifying the suitable region for the analysis enriches the competencies in measuring drought at the country level and strengthening mitigation policies for drought at the regional level. Therefore, it is essential to choose the most suitable region that can be integrated for the analysis. In this study, we have chosen the northern area of Pakistan, the selected stations of the region provide homogenous characteristics over the selected period. Further, the region has its structural importance, and most of the other areas of the country are dependent on its rainfall occurrences (*Malik et al., 2012*; *Naheed, Kazmi & Rasul, 2013*). Therefore, the selected region is important for the agriculture sectors of the country. Moreover, four drought classes (or intensities) (*i.e.*, "Extremely dry, Severely dry, Median dry, and Normal dry") are used for the current analysis. These intensities are used according to the requirements of the study; however, other drought intensities can be included in RICDPMS.

Moreover, RICDPMS is the modification of the Continuous Drought Probability Monitoring System (CDPMS) (*Pontes Filho et al., 2019*). The mentioned study used generic probabilities proposed by *Agnew (2000)* to calculate precipitation thresholds. However, the RICDPMS considers steady-state probabilities to explicitly describe the drought occurrences in the selected region. These steady-state probabilities are calculated from the data obtained by Monte Carlo Feature Selection (MCFS). Further, the probabilities attained from the steady-states method can be described as the specific drought classes with the probability of their occurrences in long-run duration and are used to calculate the thresholds for different drought intensities. The RICDPMS counts regional characteristics to enhance the capabilities for measuring drought and help early warning and drought mitigation policies to make reliable strategies at the regional level. The outcomes of the RICDPMS may improve the accuracy in actions and timely update the meteorologists and policymakers for better use of their data. The performance of RICDPMS is measured by the Brier Score (BS). The BS is a measure used to verify the binary events (yes\no). The range of the BS will persist between 0 and 1. However, if it takes the value 0, it is considered a perfect prediction. The Brier Skill Score (BSS) is used to assess the validation of the probabilistic prediction. The value of BSS remains between $-\infty$ to 1. The details and mathematical descriptions about BS and BSS are given in (*Pontes Filho et al., 2019*). The prediction is perfect when the value of BSS is close to one.

### Monte carlo feature selection (MCFS)

The MCFS technique implemented by *Niaz et al. (2020a)* is used in the current analysis for accumulating the time series data of several meteorological stations. The MCFS enables the selection of more important stations for the preliminary investigation. In this perspective, information from various stations can be determined more comprehensively. For this purpose, MCFS is applied to select the appropriate station for the varying months (6-months). For example, MCFS chooses the more appropriate station for January from six stations representing whole region characteristics. The selection of the appropriate stations is based on the values of relative importance (RIs). The mathematical details about the RI are given in *Niaz et al. (2020a)*. For instance, in January, the Chilas station takes RI values (0.2035), more significant than other selected stations. Therefore, the precipitation of this station is selected for January. The selection of the stations for February, March, April, May, and June are based on the same rationale. In this way, the precipitation of the appropriate stations is considered for the RICDPMS.

### Drought triggering precipitation thresholds based on SPI

The proposed method comprehensively models the precipitation data for various drought intensities in a homogenous climatic region. The deficit in the precipitation triggers a significant component of the drought events. The SPI has extensively been employed for assessing, monitoring, and analyzing drought incidences since the last decade (*McKee, Doesken & Kleist, 1993*). It is applied for identifying the standardized departure of the selected distribution, which is selected to model the raw data to the observed values of the precipitation. Also, it can be analyzed for distinct times intervals based on monthly data.

Moreover, the reliability of SPI has been examined considerably in climatic scenarios regarding spatial and temporal distribution. Since SPI values are standardized, therefore it is frequently considered for assessing and examining the drought, specifically for meteorological drought. Further, the calculation of SPI is simply based on the precipitation records that make it most recognized worldwide. The SPI has been utilized in various studies and several researchers have used SPI in their publications to develop new methodologies for accurate and efficient drought assessment and monitoring (*Chortaria, Karavitis & Alexandris, 2010*; *Belayneh, Adamowski & Khalil, 2016*; *Wu et al., 2021*; *Docheshmeh Gorgij et al., 2021*; *Tsesmelis et al., 2022*; *Gozalkhoo et al., 2022*). However, it is challenging for SPI to deliver speedy and comprehensive knowledge about precipitation discrepancies and drought probability in a homogenous environment (*Santos et al., 2013*; *Pontes Filho et al., 2019*; *Niaz et al., 2022*). In this research, the RICDPMS used SSP, MCFS, and copulas, to provide the drought probabilities at the regional level for various drought classes. The RICDPMS assigns the probability of having drought as the rainy season progresses by allocating the SPI at 6 months time scale (SPI-6) to drought triggering precipitation.

### Copulas functions

Knowing the dependency structure of two or more variables can help assess their predictive relationship. The Pearson correlation coefficient generally measures the linear relationship among the variables. However, this method of measuring relationships only reflects the degree of dependence while other structural dependence settlements remain unimportant in this way. Therefore, a rank correlation coefficient has been considered to determine non-linear dependence. Usually, Spearman rank correlation and Kendall's Tau are preferred for associating non-linear dependence structure. So, the use of Kendall's Tau is common because, in this way, concordant or discordant pairs' probability can be acquired directly. Also, the well-known relationship concerning the rank correlation coefficient, the copula function, is used to assess non-linear dependencies. In this perspective, Sklar's theorem is compulsory for the different aspects of varying studies in the literature (*Schweizer & Sklar 1974*; *Nelsen, 2007*; *Durante, Fernandez-Sanchez & Sempi, 2013*). Generally, the copulas' families have been categorized into four prominent groups and used in various studies for determining the dependence structure of the variables. For example, *Genest et al. (2013)* used copula models in their study and they found that the Meta-elliptical copulas were the best choice for assessing the dependence structure. *Corbella & Stretch (2013)* used Archimedean copulas for evaluating the dependence structure between storm parameters. The extreme value type copulas family was discussed in *Trutschnig, Schreyer & Fernández-Sánchez (2016)*, *Lee & Joe (2018)*. Since the reachable properties in hydrological analyses of the Archimedean copulas vary, it is very prevailing for modeling dependence structures, particularly for dependent tail structures (*Singh & Zhang, 2007*; *Serinaldi et al., 2009*; *Joe, 2014*). The use of Archimedean copulas is restricted, while characteristics of the higher-order dependency structure are desirable to examine between/among variables. This kind of higher-order characteristic in the dependent structure can be assessed using Meta-Elliptical copulas
(*Zhang et al., 2013*). Either method of moments or the Maximum Likelihood Estimation (MLE) method can be used for estimating the parameters of the copula's families. MLE method is usually preferable; however, the transformation is done to prefer the applicable method between Maximum Pseudo-Likelihood (MPL) for rank-based whereas the method of moments is limited for only one-parameters. In the current study, the data of precipitation (observed per timescale) and its sub-periods (monthly distribution) are examined by the bivariate copulas-based methodology.

The current study examines the precipitation data (observed per time scale) and its sub-periods (monthly distribution) by the bivariate copulas-based methodology. Moreover, Archimedean copulas (Gumbel and Joe) and Meta-Elliptical copulas Gaussian were tested as candidates given in Eqs. (1)–(3).

$$\text{Gumbel} \qquad \exp\left\{-\left[(-ln u_1)^{\varnothing} + (-ln u_1)^{\varnothing}\right]^{\frac{1}{\varnothing}}\right. \tag{1}$$

$$\text{Joe} \qquad 1 - [(1-u_1)^{\theta} + (1-u_2)^{\theta} - (1-u_1)^{\theta}(1-u_2)^{\theta}]^{\frac{1}{\theta}} \tag{2}$$

$$\text{Gaussian} \qquad \theta_{\rho}\left(\theta^{-1}(u_1) + \theta^{-1}(u_2)\right) \tag{3}$$

For each of the selected stations the bivariate model is established, the probability of drought occurring in the time of the rainy season is assessed by the RICDPMS. In this manner, recent precipitation records are assimilated progressively using a conditional probability for the temporal evolution of selected drought intensities. Thus, this may support anticipatory mitigation policies and be helpful to perceive early warning concerns regarding the drought incidences, raising the awareness, the civil protection authorities, and water resources management.

## RESULTS

The time-series data is compiled from six meteorological stations in the northern area of Pakistan. The climatological features of the precipitation observed in varying stations are presented in Fig. 2. The stations are selected based on the homogenous behavior of the drought classes in varying stations and indices for the specific time scale (*Ali et al., 2019a*; *2019b*; *Niaz et al., 2021a*, *2020b*, *2021d*). Further, deep classification is done using the MCFS algorithm for selecting important stations and their precipitation values, representing the whole region appropriately for the selected months. For this classification, the selection of the important stations is based on relative importance (RIs) values (see Table 1). The corresponding higher values of RIs in any station reveal that the station is considered for the analysis. For example, in January, the Chilas station has RI values (0.2035), which is higher than other selected stations. Therefore, the precipitation of this station is selected for further analysis. Further, the value 0.1445 of the Astore station for April is higher than other values of the selected stations. This selection process will continue for the other months.

Moreover, the bivariate copulas for modeling the complicated dependence structures existing between precipitation at varying time intervals. The most popular copulas

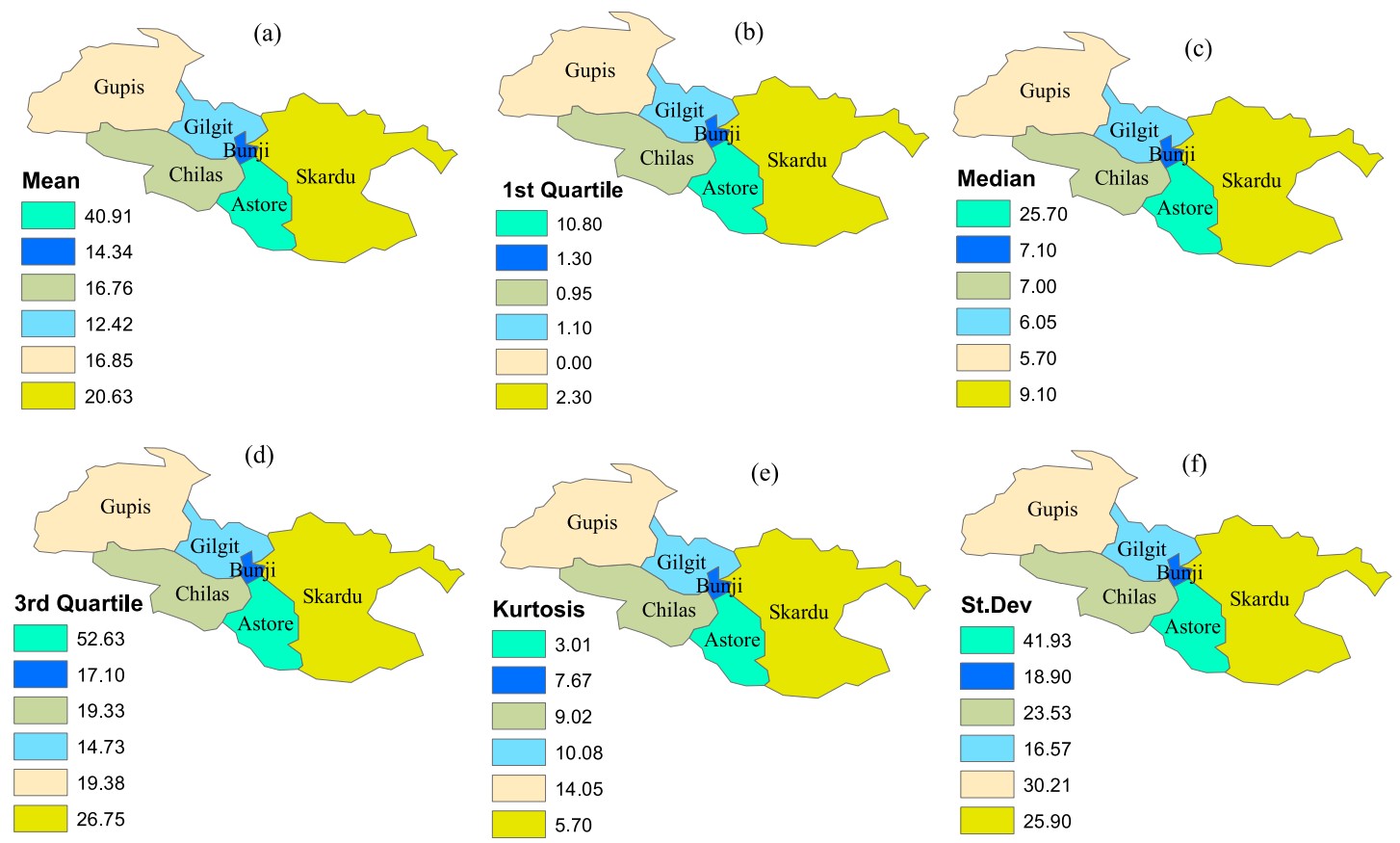

**Figure 2 The climatological fetures of the selected stations.** The varying climatological characteristics of precipitation in the selected stations. (A) Mean, (B) 1st Quartile, (C) median, (D) 3rd Quartile, (E) Kurtosis and (F) St. Dev.

| Table 1 The score for relative importance. | | | | | | |
|---|---|---|---|---|---|---|
| Stations | January | February | March | April | May | June |
| Astore | 0.1209 | 0.1423 | 0.1614 | 0.1224 | 0.0320 | **0.2225** |
| Bunji | 0.1895 | 0.1294 | 0.1032 | 0.1375 | 0.0192 | 0.1990 |
| Gupis | 0.0188 | 0.1100 | 0.0136 | **0.1445** | **0.1090** | 0.2056 |
| Chilas | **0.2035** | **0.1500** | **0.1900** | 0.0233 | 0.0232 | 0.0437 |
| Gilgit | 0.1594 | 0.1235 | 0.0116 | 0.0196 | 0.0195 | 0.1165 |
| Skardu | 0.1100 | 0.0123 | 0.1055 | 0.0300 | 0.0200 | 0.1576 |

**Note:**
The score for relative importance (RI) for each month and station. The bold font shows that the particular station for the specific month is selected for the precipitation values. The selected station becomes part of the RICDPMS.

families, are tested for the current analysis. The MPL is applied for estimating the parameters. The varying families of copulas are designated based on Akaike Information Criteria (AIC). Table 2 presents the bivariate models for the ND category. The bivariate models are chosen for each sub-period (n), for instance, the month of January ($n = 1$) and the month of May ($n = 5$). It also contains the copula's families and their parameters. The precipitation is progressively recorded in the model can be detected by the Kendall's

**Table 2 Coupled series for bivariate models.**

| $R_n$ | Family | Parameters $\varnothing$ or $\rho$ | Kendall's Tau Model | Empirical | AIC | p-value | BSS values |
|-------|--------|------|-------|-----------|-----|---------|------------|
| $n_1$ | Joe | 1.48 | 0.21 | 0.20 | −4.62 | <0.05 | 0.29 |
| $n_2$ | Gaussian | 0.54 | 0.36 | 0.34 | −10.98 | <0.05 | 0.35 |
| $n_3$ | Gumbel | 1.62 | 0.38 | 0.35 | −15.80 | <0.05 | 0.41 |
| $n_4$ | Gaussian | 0.84 | 0.63 | 0.60 | −48.36 | <0.05 | 0.67 |
| $n_5$ | Gumbel | 9.13 | 0.84 | 0.88 | −157.77 | <0.05 | 0.85 |

Note:
Each coupled series for Bivariate models, their parameters, Kendall Tau correlation according to the model and empirical, AIC, p and BSS values of Normal drought for the selected region.

Tau correlation coefficient. The structure of the dependency between $R_N$ (seasonal precipitations) and $R_n$ (sub-period precipitation) develops strongly. In this case, the $n_1$ illustrates the 21% precipitation in the time of the rainy season, while 36% precipitation can be explained by $n_2$. Further, $n_3$, $n_4$, $n_5$ explain 38%, 63% and 84% respectively. The slight disparity between the estimates of empirical Kendall's Tau and model indicates that $R_N$ and $R_n$ are properly being modeled by the copulas. The significant structure in the precipitation of the season (rainy period) and sub-period precipitation (each month within the season) are envisaged by the small p-values, which bring substantial signals against the structure of independence. Based on values of $n$ (6 months' rainy season selected for the study) the Brier Skill Score (BSS) is computed for the selected region, according to Leave-One-Out Cross-Validation (LOOCV). The computed values of BSS are used to assess the performance of the developed method. The value close to one represents that the performance of the model is enhanced. As the $n$ enhances, the capability of the RICDPMS increases as well for the prediction. The performance of RICDPMS in the selected regional setting for $n = 1$ is 0.29, while it increases to 0.85 for $n = 5$.

The year 2017 is taken as an example for the region to express the skills of RICDPMS to monitor drought continuously (see Table 3). In the first month of the rainy season, January, the precipitation is observed 3.6 mm, and the mean monthly climatological precipitation for January is 11.61 mm. The RICDPMS started to produce the risk of a drought for the available rain in the first month. So, the probability for the Normal Dry (ND) occurrence is 0.74, and the occurrence of other drought categories, including "Median, Severely, and Extremely Dry", have probabilities of 0.45, 0.40, and 0.38, respectively. Further, for February, the 2-month accumulated mean of observed precipitation, which is 27.7, already appears with a shortfall of 0.45 mm from the accumulated mean of the climatological precipitation with 28.15 mm. In March, the observed accumulated precipitation becomes 32.9 mm, which has a 23.85 mm deficit from the accumulated mean monthly precipitation. This point kept the ND probabilities high. The ND occurrence probability is 0.61. The probabilities for other drought categories, Median Dry (MD), Severely Dry (SD), and Extremely Dry (ED), are 0.18, 0.21, and 0.30, respectively. In April, the observed accumulated precipitation becomes 57.90 mm, with a 23.8 mm deficit from the accumulated mean monthly precipitation. This point kept the

**Table 3 Probabilities obtained from RICDPMS.**

|  |  | January | February | March | April | May | June |
|---|---|---|---|---|---|---|---|
| Mean Monthly Precipitation (mm) | Monthly | 11.61 | 16.54 | 28.60 | 24.95 | 26.96 | 24.80 |
|  | Accumulated | 11.61 | 28.15 | 56.75 | 81.70 | 108.66 | 133.46 |
| Observed Precipitation (mm) | Monthly | 3.6 | 24.1 | 5.2 | 25 | 29 | 16.3 |
|  | Accumulated | 3.6 | 27.7 | 32.9 | 57.90 | 86.90 | 103.2 |
| Drought category (Threshold) |  | Drought Risk |  |  |  |  |  |
| Extremely Dry (58.50) |  | 0.38 | 0.20 | 0.10 | 0.03 | 0.00 | No drought |
| Severely Dry (68.68) |  | 0.40 | 0.26 | 0.11 | 0.04 | 0.00 | No Drought |
| Median Dry (75.92) |  | 0.45 | 0.30 | 0.19 | 0.09 | 0.00 | No Drought |
| Normal Dry (125.56) |  | 0.74 | 0.42 | 0.61 | 0.69 | 0.86 | Drought |

**Note:**
Probability of Extremely Dry, Severely Dry, Median Dry, and Normal Dry events along the rainy season of 2017 (January to June) according to the RICDPMS method for the selected region.

ND probabilities high but is sufficient to suggestively decline the other categories' probabilities as they look closer to their precipitation thresholds. Further, the ND probability is 0.69 which is higher than other drought categories. In May, the ED, SD, MD, conditions were already out of possibility as the accumulated precipitation by this month, 86.90 mm, was already higher than their thresholds of 58.50, 68.68, and 75.92, respectively. When June is ended, the accumulated mean precipitation for 2017 was 103.2 mm, below 133.46 mm, which is the climatological mean monthly accumulated precipitation for the period and not enough to meet the 125.56 mm that indicates the ND threshold. Therefore, RICDPMS permits decision-makers for recognizing the rising risk of drought in that year and what category it represents. For example, ND state (intensity) in March has a drought risk with 0.61 (probability). Therefore, to mitigate the potential of the negative outcomes of Normal drought, the anticipated measures could be implemented accordingly.

## DISCUSSION

The bivariate copulas are used with their most famous families to fulfill the study's objective. The parameters estimation and class of the copulas are selected based on MPL and AIC accordingly. The last step refers to monitoring the progression of the drought probability at the time of the rainy season in the selected region. The RICDPMS establishes for computing drought-triggering precipitation thresholds for varying drought intensities given by the SPI (*Santos et al., 2013*; *Pontes Filho et al., 2019*). *Pontes Filho et al. (2019)* developed a new method named as Continuous Drought Probability Monitoring System (CDPMS) that monitors drought occurrences. The CDPMS translates the probability of drought incidents into user-friendly mathematical transformations. The CDPMS was validated at the rain gauge level in Portugal, but the drought is usually extended in large areas. The CDPMS has utilized generic probabilities of *Agnew (2000)*. Further, the precipitation thresholds were calculated by CDPMS based on the Gamma distribution only. Moreover, *Niaz, Almazah & Hussain (2021)* proposed Modified CDPMS (MCDPMS) for drought monitoring. The MCDPMS has more significance

than CDPMS because it used appropriate probability distributions according to the precipitation nature of the selected stations. It also used steady-state probabilities for the computation of precipitation advances. The bivariate copulas were also used for the development of MCDPMS. Moreover, the MCDPMS was stations specified and validated on 12 meteorological stations of the province of Punjab, Pakistan.

Recently, Niaz et al. (2022) proposed a new method that monitored drought at the regional level. They used six stations in the northern areas of Pakistan. The study had utilized spatial information about precipitation advances from various stations. It selected the precipitation values from varying stations based on the weighting scheme. However, these stations were homogenous (Ali et al., 2019a; 2019b; Niaz et al., 2020b; 2021d), and therefore, it seems counterproductive to study all stations for choosing appropriate values. Furthermore, it was observed that the weighting scheme assigned higher weights for specific drought classes. Thus, those selected classes contained higher precipitation values for the region and were used to determine drought severity for the region. Therefore, due to high precipitation in selected months, several drought classes went out of option in most of the months as compared to their precipitation thresholds. These issues underpin the new methodology that can consider only representative stations. Hence, the current study develops a new method, RICDPMS that monitors only important stations for the precipitation advances. The development of the RICDPMS is based on the MCFS, steady-state probabilities, and various copulas functions. The MCFS is employed for the selection of more important stations. The MCFS mainly helps to minimize the time and resources. The use of MCFS makes RICDPMS more efficient than CDPMS (Pontes Filho et al., 2019), MCDPMS (Niaz, Almazah & Hussain, 2021), and Regional spatially agglomerative continuous drought probability monitoring system (RSACDPMS) (Niaz et al., 2022) for drought monitoring specifically in the selected region. Further, the SSP are utilized for calculating the regional thresholds of the precipitation for different drought intensities. The RICDPMS also considers the bivariate copulas for modeling the complicated dependence structures between precipitation at varying time intervals. Hence, RICDPMS assesses the spatial-temporal drought variability in the selected region and provides immediate information about precipitation advances in the selected region. It also monitors the regional drought with varying drought intensities. The RICDPMS can be utilized for drought monitoring and mitigation policies. Further, the outcomes obtained in this study can only be related to the current conditions of the application site because upcoming possible climate conditions could invalidate the extrapolations based on analysis of such long data sets. Further, the embodiment of daily precipitation data such as in the RICDPMS can enhance drought monitoring in the selected region.

## CONCLUSION

Drought is one of the complicated natural calamities in terms of disrupting the natural life of the people directly. Moreover, continuing lack of precipitation causes pernicious results on society and the economy. Therefore, meticulous measures of drought monitoring are compulsory to identify drought incidences that perform more comprehensively and credulously for early warning and mitigation policies. Moreover,

accurate and comprehensive information at the regional level regarding drought characteristics will increase analysts' and policymakers' competencies and time efficiency to make their cost-effective plans and mitigation policies. For this purpose, we propose a method known as the Regional Intensive Continuous Drought Probability Monitoring System (RICDPMS) for monitoring drought characteristics more comprehensively and expeditiously in the selected region. The proposed method is accomplished by using MCFS, SSP, and copula-based bivariate analysis for different drought intensities. The data gathered from six meteorological stations in the northern zones of Pakistan is utilized to validate the proposed methodology. As a result, we found that the RICDPMS monitors the drought comprehensively and expeditiously in the region and provides a comprehensive approach for evaluating drought with different drought intensities in the region. In summary, the RCDPMS may use for drought monitoring and mitigation policies.

### Funding

Mohammed M. Almazah received funding for this work under grant number RGP. 1/10/43 from the Deanship of Scientific Research at King Khalid University. The funders had no role in study design, data collection and analysis, decision to publish, or preparation of the manuscript.

### Grant Disclosures

The following grant information was disclosed by the authors:
King Khalid University: RGP. 1/10/43.

### Competing Interests

The authors declare that they have no competing interests.

### Author Contributions

- Rizwan Niaz conceived and designed the experiments, performed the experiments, analyzed the data, prepared figures and/or tables, authored or reviewed drafts of the paper, and approved the final draft.
- Mohammed M. A. Almazah conceived and designed the experiments, analyzed the data, prepared figures and/or tables, and approved the final draft.
- Ijaz Hussain conceived and designed the experiments, performed the experiments, analyzed the data, prepared figures and/or tables, supervision, and approved the final draft.
- Muhammad Faisal conceived and designed the experiments, performed the experiments, prepared figures and/or tables, authored or reviewed drafts of the paper, and approved the final draft.
- A. Y. Al-Rezami conceived and designed the experiments, performed the experiments, prepared figures and/or tables, and approved the final draft.

- Mohammed A. Naser conceived and designed the experiments, performed the experiments, prepared figures and/or tables, and approved the final draft.

## Data Availability
The raw data is available in the Supplemental File.

## Supplemental Information
Supplemental information for this article can be found online at http://dx.doi.org/10.7717/peerj.13377#supplemental-information.

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
