# Peer review of "A new comprehensive approach for regional drought monitoring"

_PeerJ, doi:10.7717/peerj.13377_

## Round 0.1 · original submission · Major Revisions

Please carefully consider all the comments made by the 2 reviewers and respond, accordingly changing the sections that need them (Abstract, Results and Discussion), and improving the clarity of the figures.

Reviewer 1 ·

Basic reporting

no comment

Experimental design

The authors in this study proposed a Regional Intensive Continuous Drought Probability Monitoring System (RICDPMS) method to obtain quick and comprehensive information regarding the drought probability and the temporal evolution of the droughts at the regional level, which was accomplished by using Monte Carlo Feature Selection (MCFS), steady-state probabilities, and copula-based bivariate analysis for different drought intensities.

Validity of the findings

The paper showed through its application in northern area of Pakistan that the RICDPMS could monitor the region's drought and provide a better quantitative way to analyze drought with varying drought intensities in the region. Therefore, this study has a certain meaning in practical applications and it is recommend accepting this manuscript.

Additional comments

Yet I still have several specific concerns for authors to improve their manuscripts.
1) The keywords included “drought risk monitoring”. However, there was no description of drought risk in the introduction and methods. Can you add relevant research in the introduction and explain the specific calculation method of drought risk in this study?
2) Lines 90-93 introduced that the precipitation in the study area was concentrated from January to June. Therefore, the drought in this period should be the lowest frequency and lowest intensity throughout the year. However, this paper selected only the rainy season for drought monitoring and analysis, please explain why, or does it have more practical significance and value to extend the research time series to the whole year?
3) In this paper, the MCFS method was used to represent the data of a certain station in the region to the entire research area. The intention was to reduce time and capital investment while maintaining the accuracy of the results. However, whether the analysis results of the comprehensive data obtained after using this method were the same or similar as the results obtained by the complete data using the traditional method? I hope the authors to add this comparison to verify the reliability and accuracy of using this method.
4) Please explain the reasons for choosing SPI-6 instead of other time scales as the drought indicator.
5) The title of Figure 1 is “Geographical locations of the six selected stations of the region of Pakistan”. However, the specific location of the meteorological stations were not marked, please add it to the map.
6) Line 119: Please check the title format and delete extra punctuation.
7) Line 226-228: Please check whether the sentence expression is correct.
8) Line 250: Please elaborate on the meaning of "climatological precipitation". Is it the average rainfall for many years or the rainfall threshold or something else?

·

Basic reporting

The authors proposed a Regional Intensive Continuous Drought Probability Monitoring System (RICDPMS) method to obtain quick and comprehensive information regarding the drought probability and the temporal evolution of the droughts at the regional level, as a study of the Pakistan.
The topic is of great regional importance and interesting results are shown in the paper, which could help readers to monitor the drought in a regional level. However, a few things need to be clarified before being considered for publishing. My further comments below contain some suggestions for extensions to the current manuscript.

Experimental design

Your manuscript structure needs clearer: “3. Results. 4. Discussion.”

Validity of the findings

no comment

Additional comments

1. The abstract needs improvement. You should be sure that the abstract provides a concise problem statement and briefly explains the main findings and their significance.
2. The figures needs more detail in this manuscript. A figure is worth a thousand words. I suggest that you improve the figures in this manuscript to provide more visual information of the drought monitoring.
3. RICDPMS is the modification of the CDPMS (Pontes Filho et al., 2019). What are the advantages and improvements in RICDPMS? I suggest that you add the discussion to provide more justification for your study, specifically, you should compare the drought monitoring effects of the two methods.

---

## Round 0.2 · Minor Revisions

You have addressed all the comments made by both reviewers and there are additional comments on the technical aspect of your manuscript. However, there are a few instances in which the English text needs revisions. For example, line 2 in the Abstract reads "...researchers have been used SPI..." which is not grammatically correct. Pleaser revise to "...researchers have used SPI..." or "...researchers have been using SPI".
Articles are missing in some sentences.

I recommend close inspection of the full manuscript to detect potential grammatical/typographical errors before I can Accept it.

Reviewer 1 ·

Basic reporting

ok

Experimental design

ok

Validity of the findings

ok

Additional comments

The authors have explained the questions I raised in detail and revised the unreasonable parts accordingly. Now the structure of the article is clarity and the application validated that the Regional Intensive Continuous Drought Probability Monitoring System has a certain meaning in practical applications. Therefore, it is recommend accepting this manuscript in its current version.

·

Basic reporting

no comment

Experimental design

no comment

Validity of the findings

no comment

Additional comments

The authors have addressed all my comments. I do not have any further suggestion.

---

## Round 0.3 · Minor Revisions

I reiterate that while the science is sound and the manuscript could be accepted, the English text still needs to be revised. Phrases such as the second line in the Abstract are incorrect.

---

## Round 0.4 · accepted · Accept

Thank you for modifying the Abstract as I had recommended.